# Reconstruction-guided attention improves the robustness and shape processing of neural networks

**Seoyoung Ahn**[1], **Hossein Adeli**[1], **Gregory J. Zelinsky**[1,2]
[1]Stony Brook University, Department of Psychology, Stony Brook, NY,
[2]Stony Brook University, Department of Computer Science, Stony Brook, NY
`seoyoung.ahn@stonybrook.edu`

## Abstract

Many visual phenomena suggest that humans use top-down generative or reconstructive processes to create visual percepts (e.g., imagery, object completion, pareidolia), but little is known about the role reconstruction plays in robust object recognition. We built an iterative encoder-decoder network that generates an object reconstruction and used it as top-down attentional feedback to route the most relevant spatial and feature information to feed-forward object recognition processes. We tested this model using the challenging out-of-distribution digit recognition dataset, MNIST-C, where 15 different types of transformation and corruption are applied to handwritten digit images. Our model showed strong generalization performance against various image perturbations, on average outperforming all other models including feedforward CNNs and adversarially trained networks. Our model is particularly robust to blur, noise, and occlusion corruptions, where shape perception plays an important role. Ablation studies further reveal two complementary roles of spatial and feature-based attention in robust object recognition, with the former largely consistent with spatial masking benefits in the attention literature (the reconstruction serves as a mask) and the latter mainly contributing to the model's inference speed (i.e., number of time steps to reach a certain confidence threshold) by reducing the space of possible object hypotheses. We also observed that the model sometimes hallucinates a non-existing pattern out of noise, leading to highly interpretable human-like errors. Our study shows that modeling reconstruction-based feedback endows AI systems with a powerful attention mechanism, which can help us understand the role of generating perception in human visual processing.

## 1 Introduction

One of the hallmarks of human visual recognition is its robustness against various types of noise and corruption (Vogels & Biederman, 2002; Avidan et al., 2002). Despite the remarkable success of convolutional neural networks (CNNs) and their applications for various image classification tasks, CNNs can still be highly vulnerable to small changes in object appearances even when the changes are almost imperceptible to humans (Szegedy et al., 2013; Dodge & Karam, 2017; but see Geirhos et al., 2021 for CNNs reaching the human-level performance in recognizing distorted images). Recent studies also revealed that CNNs exploit fundamentally different feature representations than those used by humans (e.g., relying on texture rather than shape, Baker et al., 2018; Geirhos et al., 2018), suggesting that there is still a gap in building an object recognition system with human-like robustness.

4th Workshop on Shared Visual Representations in Human and Machine Visual Intelligence (SVRHM) at the Neural Information Processing Systems (NeurIPS) conference 2022. New Orleans.

How humans can robustly recognize objects despite their tremendous complexity and ambiguity in nature has been a long-standing question in disciplines ranging from cognitive science (Biederman, 1987; Tarr & Pinker, 1990) and neuroscience (Plaut & Farah, 1990; Rolls, 1994) to computer vision (Marr, 1982; Ullman, 1989). Our approach is to apply generative modeling to this question. A generative model of perception suggests that top-down generative feedback is what enables the brain's robustness and better generalization to novel situations (Dayan et al., 1995; Yuille & Kersten, 2006; de Lange et al., 2018). According to this theory, the brain constantly attempts to generate percepts consistent with our hypotheses about the world, a process referred to as "prediction" in predictive coding theory (Clark, 2013) or "synthesis" in analysis-by-synthesis theory (Yuille & Kersten, 2006). Although it is still debated whether human visual cortex encodes an explicit generative model that evaluates the likelihood of all possible states of the world (DiCarlo et al., 2021; Gershman, 2019), the idea that the brain actively reconstructs visual percepts using its internal knowledge is generally supported by evidence from many visual phenomena such as imagery, perceptual filling-in and completion, and hallucination (e.g., pareidolia). However, it is still largely unknown how this top-down reconstruction is integrated with bottom-up information and why it contributes to robust visual recognition.

Here we propose that top-down object reconstruction can serve as an effective attentional template to bias neural competition in favor of the most plausible or recognizable object. Prior work shows that the content of short-term memory, keeping track of an object's location and properties, fuels the biasing signals of attention (Desimone & Duncan, 1995; Beck & Kastner, 2009; Bundesen et al., 2011). Inspired by this, we designed a neural network that generates an object reconstruction—a visualized prediction about the possible appearance of an object—and uses it as top-down attentional feedback to dynamically route object-relevant spatial regions and features. Recent modeling work of generative vision by Yildirim et al. (2020) suggested that the visual recognition pathway, the brain's ventral stream, is trained to reverse-engineer the generative process and is used to infer the underlying cause of the retinal input. Contrary to this work, we assume that the generative process (the object reconstruction process in our case) can also be used during inference as top-down attentional bias potentially via a dorsal pathway including the parietal cortex, which is known to be involved in top-down controlled attentional selection (Vossel et al., 2014; Adeli et al., 2021). Our model also adopts an object-centric approach, where an input object is first encoded to highly compressed and abstract representations (referred to as "slots" in object-centric generative models; Greff et al., 2020; Locatello et al., 2020) and information about an object (e.g., location, shape) can later be decoded to accomplish the task at hand. While the object-centric approach is known to improve the model's generalization performance (Dittadi et al., 2021), this has been primarily explored in the context of pretraining or designing auxiliary loss (Rasmus et al., 2015; Sabour et al., 2017; Chen et al., 2020; He et al., 2021). However, their potential application to inference as attentional bias was never studied.

In this work, we aim to explore whether object reconstruction-based feedback can be used actively during inference and observe the role of this attention mechanism in the artificial vision systems. We demonstrate that the proposed object reconstruction-guided attention yields the most robust classification performance in digit recognition under various types of corruptions (MNIST-C; Mu & Gilmer, 2019). This benefit was more pronounced under certain corruptions (noise, blur, and occlusion), where greater global shape processing is required. The model also yields qualitative behavioral correspondence with humans measured by the inference speed and the types of errors made. Future studies will scale to more naturalistic images and complex scenes with multiple objects.

## 2  Model Architecture and Training

Fig. 1 shows the model pipeline. Our model has an encoder-decoder architecture, where the outputs of a convolutional encoder are grouped into a fixed set of neural "slots" (Greff et al., 2020; called "capsules" in CapsNet, Sabour et al., 2017) to represent each entity (e.g., objects or object parts). The model includes eight-dimensional slots for encoding object features (**feature capsules**) and sixteen-dimensional slots for encoding objects (**object capsules**). Our model is trained to both read out the classification scores and generate the reconstruction of an input using the object capsules. The main novelty of our approach is in using object reconstructions to form top-down spatial and feature-based attentional biases during inference. We used margin loss (Sabour et al., 2017) for classification loss and mean squared error (squared L2 norm) between the input and reconstructed image for reconstruction loss. Appendix A gives detailed model training settings.

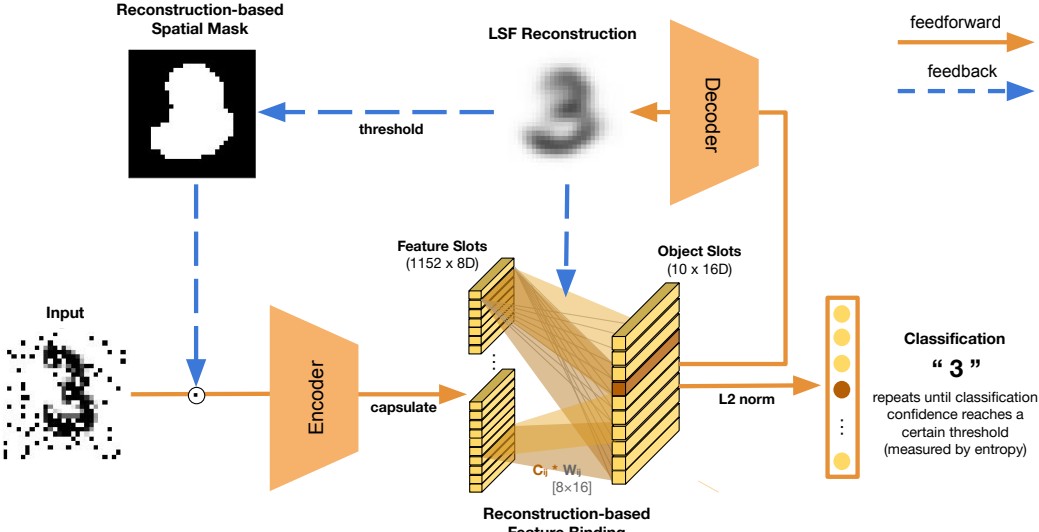

Figure 1: Model architecture. Our model uses object reconstruction as top-down attentional feedback and routes the most relevant visual information to feed-forward object recognition processes. This reconstruction-guided attention operates at two levels: global projection that suppresses responses to irrelevant spatial regions outside of the most likely object's reconstruction (Reconstruction-based Spatial Mask) and local recurrences that control binding strengths from features to objects in favor of forming a more veridical object reconstruction (Reconstruction-based Feature Binding). The model repeats its global iteration until the classification confidence reaches a certain threshold. The number of local recurrent steps is fixed to 3. See the main text for details.

## 2.1 Reconstruction-guided Attention Mechanism

Here we introduce two mechanisms for object reconstruction to serve as top-down feedback, one consisting of a long-range global feedback connection that serves as a spatial mask and the other, a local recurrence between adjacent layers that realizes a feature-biasing function.

**Reconstruction-based Spatial Mask** performs the long known role of spatial filter of attention, a mechanism that allows inputs within an attended region to pass through but attenuates inputs falling outside its extent (Posner, 1980). The model generates a boolean mask from thresholding the most likely object's reconstruction at 0.1 (the original pixel intensities range from 0-1) and then normalizes the intensity values within the input area selected by the boolean mask (See Fig. 2A for illustration). This spatial mask therefore constrains spatial attention to be in the shape of an object, which is also believed to happen in humans (Duncan, 1984). This global iteration stops when the model's classification confidence level reaches a certain threshold, measured by the entropy over the softmax distribution of classification scores (similar to the methods used in Spoerer et al., 2020). We chose an entropy threshold of 0.6 by using grid-search method. If this threshold is never reached within a fixed number of steps (T=5), then the prediction from the final time step is used for classification.

**Reconstruction-based Feature Binding** dynamically modulates the routing coefficients from the feature capsules to the object capsules. The routing coefficients reflect binding strengths between features and objects ($C_{ij} \in \mathbb{R}^{228 \times 10}$) that are multiplied by the original weights between two capsule layers ($C_{ij} * W_{ij}$ when $W_{ij} \in \mathbb{R}^{8 \times 16}$). Only the weights are updated through error backpropagation, the routing coefficients are computed dynamically during inference over three iterations. Many methods have been explored in the literature for determining the routing coefficients (Sabour et al., 2017; Hinton et al., 2018; Zhao et al., 2019; Tsai et al., 2020), that rely largely on representational similarity between features and object capsules. Our method extends those approaches by also taking into account reconstruction of each hypothesized object to suppress routing of features to object slots with low reconstruction scores (an inverse of reconstruction error). As shown in Fig. 2B, this algorithm results in routing coefficients sharpening over iterations and becoming more focused on object features that best match with the input (conceptually similar to "explain-away" behavior, Clark, 2013). Routing algorithm is provided in Appendix C.

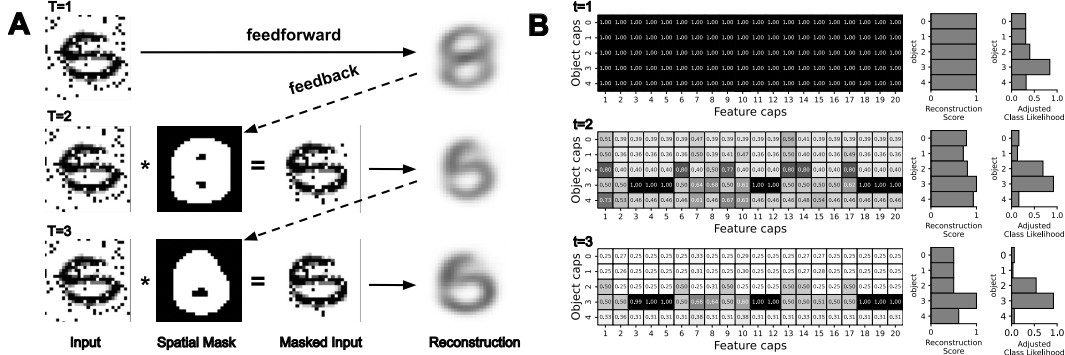

Figure 2: Step-wise visualizations of reconstruction-based attention. **A**: Object reconstructions and resulting spatial masks at each global step. The spatial mask limits the focus of attention within the shape of the most likely object. In this example, the model's object reconstruction from the most likely object hypothesis changes from 8 to 5 through 3 steps (when groundtruth class is 5) **B**: Left column shows the routing coefficient matrix between object capsules (rows) and feature capsules (columns) at each local step. The routing coefficients are all ones at the initial step (dark color on top) but get suppressed (become lighter) over iterations if they are connected to the object features that lead to lower reconstruction accuracy. In this example, the coefficient matrix becomes sparser with each iteration focusing on the features of digit 3 (darker line along the fourth row), the hypothesis that yields the highest reconstruction score as shown in the middle column. The rightmost column is the classification score (class likelihood) adjusted by the reconstruction score. For illustration purposes, only 5 object capsules and 20 feature capsules are shown. See Appendix B for the full figure.

## 2.2 Training and Testing

We trained the model on MNIST and tested its generalization performance on MNIST-C (Mu & Gilmer, 2019). This dataset has 15 different types of corruptions including affine transformation, noise, blur, occlusion, etc (**MNIST-C** in Tab. 1). We also report results separately on a subset of the dataset (**MNIST-C-shape** in Tab. 1), which includes those corruptions relating to object shape so as to best evaluate the role of shape reconstruction in our model performance (Geirhos et al., 2021). This subset includes corruptions that maintain the overall shape of the object but change its local parts or texture due to the addition of noise, blur, or occlusion. See Appendix D for example images. Note that our model is only trained using clean MNIST digits and therefore was never exposed to any of the corruption types used for testing. We explored what degree of precision is required for reconstruction and trained our models to either reconstruct the full-spectrum, low-frequency, and high-frequency versions of the input, with the latter two created by applying low-pass and high-pass filters on the input images (Appendix E).

## 3 Results

To test whether the use of reconstruction-guided attention improves robustness, we built the model that uses object reconstructions as a top-down attentional bias and evaluated the model's performance in recognizing digits under various corruptions. We considered two types of convolutional backbones for encoders in our model: a shallow 2-layer convolutional neural network (2 Conv) and a deep 18-layer residual network (Resnet-18; He et al., 2016). We also implemented three baseline models: two are simple CNNs comprised of the same backbones as above followed by fully-connected layers and the third is the original CapsNet model (Sabour et al., 2017). Average performance from 5 random initializations is reported for the models we implemented. The best performance from other baselines (e.g., generative and adversarially trained models; see Tab. 1 for details) are taken from the original benchmarking paper (Mu & Gilmer, 2019). The source code is available here[1].

---

[1] https://github.com/ahnchive/recon-mnist

## 3.1 Model Comparison

As shown in Tab. 1, our model with a Resnet-18 encoder achieves the best robustness as measured by both MNIST-C and MNIST-C-shape. We discovered that even the use of a low-resolution object reconstruction (low-spatial frequency image reconstruction) resulted in strong recognition robustness and thus we are reporting the results from this version. See Appendix E for results from the model trained with high and full-spectrum frequency image reconstructions.

The original benchmarking study (Mu & Gilmer, 2019) showed that even the models specially designed and trained for generalization (e.g., adversarially trained models) did not achieve better performance than simple CNNs on this task. On the contrary, our model outperformed simple CNNs by a wide margin ($> 4\%$), performing particularly well on MNIST-C-shape. We also observed that increasing the capacity of the encoder from a 2-layer convolutional to an 18-layer residual network led to over-fitting by a simple CNN, resulting in degradation of its performance from 88.94% to 81.94% on MNIST-C. Our model did not suffer this problem and achieved higher performance with the higher capacity encoder.

Table 1: Model comparison results. The average performance and the standard deviations from 5 runs are reported for **Our**, **CNN**, and **CapsNet** models (see text for details). The best model performance for other baselines are from Mu & Gilmer (2019). **ABS**: a generative model (Schott et al., 2018). **PGD1**: CNN trained against PGD adversarial noise (Madry et al., 2018). **PGD2/GAN**: another CNN trained against PGD/GAN adversarial noise (Wang et al., 2018).

| Dataset | Our | | CNN | | CapsNet | ABS | PGD1 | PGD2 | GAN |
|---|---|---|---|---|---|---|---|---|---|
| | Resnet-18 | 2 Conv | Resnet-18 | 2 Conv | | | | | |
| MNIST-C | **91.84 (0.67)** | 88.32 (0.34) | 81.94 (0.43) | 88.94 (0.64) | 75.14 (1.00) | 82.46 | 80.06 | 78.86 | 81.14 |
| MNIST-C-shape | **96.24 (0.74)** | 92.82 (0.90) | 86.43 (0.44) | 91.80 (0.88) | 81.87 (0.55) | 88.69 | 83.43 | 81.38 | 83.53 |

Fig. 3A and B show comparisons between Our vs CNN on each of the corruption types from MNIST-C-shape. Our model generally outperforms the CNN baselines, and particularly well under the fog corruption. Our model completes recognition in 1.4 steps on average (measured as the number of global timesteps required to reach a confidence threshold), but took significantly longer ($> 2$ steps) under fog corruption, suggesting that feedback processing is helpful for robust object recognition under challenging situations (Wyatte et al., 2012; Kar et al., 2019; Rajaei et al., 2019).

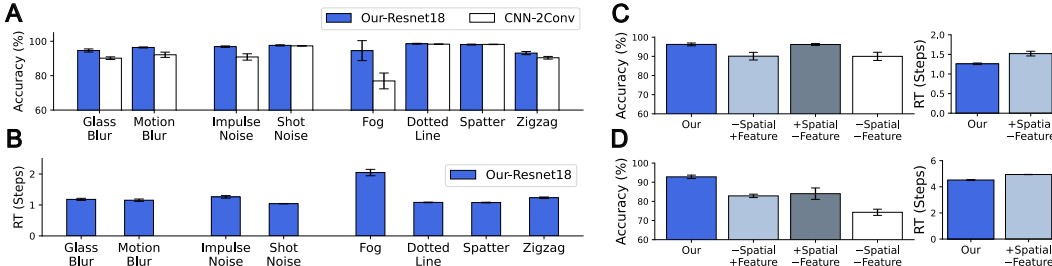

Figure 3: Detailed model evaluations. **A:** Our vs CNN accuracy comparison on corruption types from MNIST-C-shape. **B:** Our model reaction time (RT; number of global time steps required to reach a confidence threshold). **C, D:** Our model performance when its reconstruction-based spatial mask and feature binding were removed (C: our model with a Resnet-18 encoder, D: our model with a 2-layer convolutional encoder). Averages are reported from 5 different runs. Error bars are SDs.

## 3.2 Ablation Studies

We examined the unique contribution of two types of reconstruction-guided attention by ablating each component and testing the model performance. Fig. 3C shows that object reconstruction is particularly useful for spatial masking, with performance dropping $> 6\%$ when the spatial masking is removed. Ablating feature binding on the other hand did not degrade the model's prediction accuracy but significantly increased the model RT (the number of global steps the model takes to recognize

digits with confidence). Feature binding was shown to have an impact on the model's robustness when the network encoder had a lower capacity (e.g. 2 Conv. layers; Fig. 3D).

### 3.3 Qualitative Observation

Fig. 4 provides qualitative evaluations of model behavior. We observe that the model sometimes alternates its prediction (Fig. 4A) and takes longer to converge on one hypothesis, which seems to reflect recognition difficulty and ambiguity. The model also often hallucinates a non-existing pattern out of noise, leading to highly interpretable human-like errors (e.g., reconstruct 7 instead of 1 when a zigzag line crosses the center, Fig. 4B top). Our future work will collect actual human recognition responses through psychophysical experiments and examine if our model yields higher behavioral correspondence with humans than does the CNN baseline.

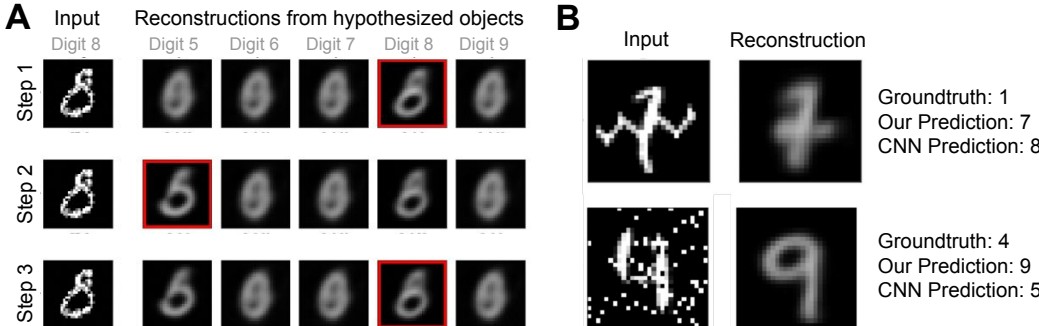

Figure 4: Qualitative model behavior. **A:** Our model takes longer to converge on one hypothesis when the input is ambiguous. The first column shows the original input, and the rest are the reconstructions from hypothesized objects. The most likely object hypothesis at each step is indicated with a red square. The model alternates its best guesses between 8 and 5 through steps. **B:** Our model produces more human-interpretable errors than does the CNN baseline. First column: original input images. Second column: model's final reconstruction. The digit in the first row was reconstructed as 7 when the groundtruth is 1 to account for the zigzag line. Similarly, the digit in the second row was reconstructed as 9 when the groundtruth is 4 because of noise speckles connecting two vertical lines.

## 4 Discussion

Human vision operates as if it has a generative engine to construct and simulate the visual world (Gershman, 2019; Battaglia et al., 2013). Here we tested whether using an object reconstruction as a top-down attentional bias can improve recognition robustness under various corruptions. Our model uses object reconstruction iteratively to route the most relevant spatial and feature information to feed-forward object recognition processes. We found that even the use of a low-resolution object reconstruction improved recognition robustness, results consistent with prior findings that our brain uses low spatial frequency information (e.g., the global shape and structure of a scene) as top-down feedback (Bullier, 2001; Bar, 2003; Bar et al., 2006). Our model is particularly robust to blur, noise, and occlusion corruptions, where processing the correct shape of an object plays an important role. Ablation study revealed different roles played by these spatial and feature biases. A reconstruction-based spatial mask enables selective attention to a region defined by the hypothesized object's shape (Scholl, 2001; Chen, 2012), leading to more robust classification. In comparison, reconstruction-based feature binding contributes more to inference efficiency (number of iterations needed by the model to make its classification) by prioritizing object features in explaining the input (Maunsell & Treue, 2006; Greff et al., 2020) rather than the model's final class prediction (unless the network encoder has a lower capacity, e.g. 2 Conv. layers). Future work will apply this model to other datasets and tasks in order to gain a better understanding of the potential role of object reconstruction in creating robust visual perception more broadly.

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

# Appendix

## A   Model implementation and hyperparameter setting

All model parameters are updated using Adam optimization (Kingma & Ba, 2014) with a mini-batch size of 128 and a learning rate exponentially decayed by 0.96 every training epoch from 0.1. Model training was terminated based on the results from the validation dataset (10% of randomly selected training dataset) and early stopped if there is no improvement in the validation accuracy after 20 epochs. The model definition for Resnet-18 and 2 Conv encoder are taken from here and here. The reconstruction decoder is 3 fully-connected dense layers All the experiments in this paper were implemented in the PyTorch framework on a single GPU with 24 GB of RAM and an NVIDIA GeForce RTX 3090. Detailed model specifications and total number of trainable parameters are reported below.

| Model type | with Resnet-18 Encoder | with 2 Conv Encoder |
|---|---|---|
| # of feature capsule | 288 | 1152 |
| feature capsule dimension | 8 | 8 |
| # of object capsule | 10 | 10 |
| object capsule dimension | 16 | 16 |
| Total trainable params | 4581296 | 2904720 |

## B   Step-wise visualization of reconstruction-guided feature binding process

The left column shows the routing coefficient matrix between all object capsules (rows) and the first forty feature capsules (columns) at each local step. The routing coefficients are all ones at the initial step (dark color on top) but get suppressed (become lighter) over iterations if they are connected to the object features that lead to lower reconstruction accuracy. The feature binding process results in routing coefficients sharpening over iterations and becoming more focused on object features that best match the input. **Left horizontal bars:** classification scores for each object capsule before feature routing is applied. **Middle horizontal bars:** reconstruction scores (an inverse of reconstruction error) from the current object capsules. **Right horizontal bars:** classification scores for each object capsule after feature routing is applied. Best viewed with zoom in PDF.

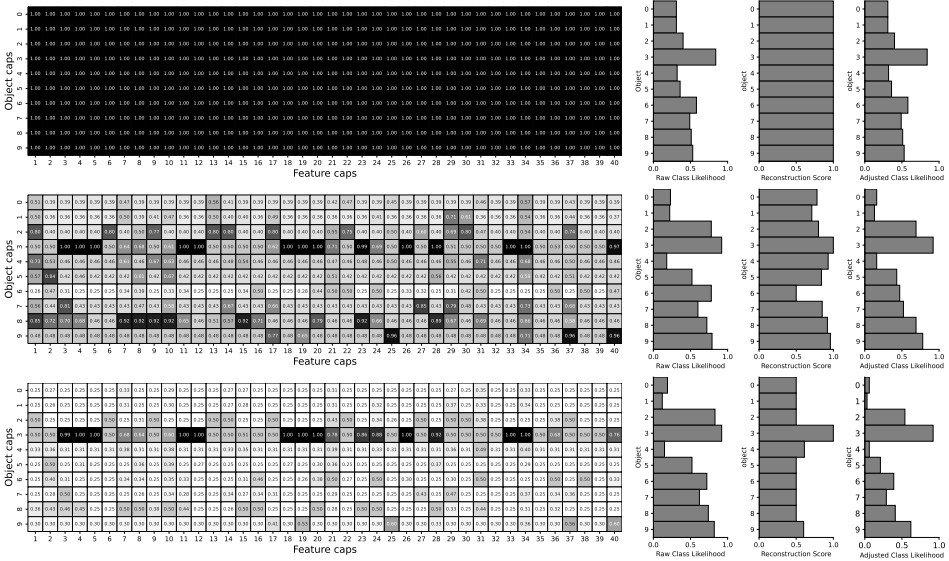

## C    Pseudocode for reconstruction-guided feature binding algorithm

In the original CapsNet (Sabour et al., 2017), the binding coefficients between feature-level capsules $i$ and object-level capsules $j$ are modulated by their representational similarity (dot product between the object-level capsules and the predictions from each feature-level capsule; Line 6 in Pseudocode 1). In our model, these binding coefficients are further adjusted to make the features that are connected to an object with high reconstruction error become more suppressed than the others.

---

**Pseudocode 1** Reconstruction-guided feature binding

---

1: **procedure** CAPSULE($p_t$)
2:      For all object capsule $j$ and primary capsule $i$ initialize $c_t^{ij}$ to one
3:      **while** $routings = 1, 2, \ldots$ **do**
4:          For all object capsule $j$ compute prediction from all primary capsule $i$: $\hat{p}_t^{j|i} \leftarrow W_t^{ij} p_t^i$
5:          For all object capsule $j$: $d_t^j \leftarrow squash(\sum_i c_t^{ij} \hat{p}_t^{j|i})$      \\ squash function in Eq. 1
6:          For all object capsule $j$ and primary capsule $i$: $b_t^{ij} \leftarrow a_t^{ij} + \hat{p}_t^{j|i} \cdot d_t^j$
7:          $a_t^{ij} \leftarrow max(maxmin(a_t^{ij}, dim = j), 0.5)$                      \\ maxmin function, Eq.2
8:          For all object capsule $j$: $recon_t^j \leftarrow Decoder(d_t^j)$
9:          For all object reconstruction $j$: $r_t^j \leftarrow -MSE(image_t^j, recon_t^j)$
10:         For all primary capsule $i$: $r_t^{ij} \leftarrow Repeat(r_t^j)$
11:         $r_t^{ij} \leftarrow max(maxmin(r_t^{ij}, dim = j), 0.5)$                      \\ maxmin function, Eq.2
12:         $c_t^{ij} \leftarrow a_t^{ij} \cdot r_t^{ij}$
13:     **end while**
14:     For all object capsule $j$: $d_t^j \leftarrow squash(\sum_i c_t^{ij} \hat{p}_t^{j|i})$
15: **end procedure**

---

### C.1    Squash function

Like the original CapsNet (Sabour et al., 2017), we used a vector implementation of capsules where the capsule vector's length represents the presence of an instance of each object class. To make this value ranges from 0 to 1 (i.e., 0 for absence and 1 for presence), the following nonlinear squash function was used.

$$d_t^j = \frac{\|v_t^j\|^2}{1 + \|v_t^j\|^2} \frac{v_t^j}{\|v_t^j\|} \tag{1}$$

### C.2    Max-min normalization

$$c_t^{ij} = lb + (ub - lb) \frac{c_t^{ij} - min(c_t^{ij})}{max(c_t^{ij}) - min(c_t^{ij})} \tag{2}$$

# D   Examples of MNIST-C testing dataset

The model was trained to reconstruct the clean MNIST training images and tested on the challenging out-of-distribution digit recognition task, MNIST-C, where 15 different types of corruption (e.g., noise, blur, occlusion, affine transformation, etc) are applied to handwritten digit images (Mu & Gilmer, 2019). We also report results on a subset of this dataset (**MNIST-C-shape**), which only includes noise, blur, or occlusion, the corruptions relating to object shape. Luminance was inverted for visualization.

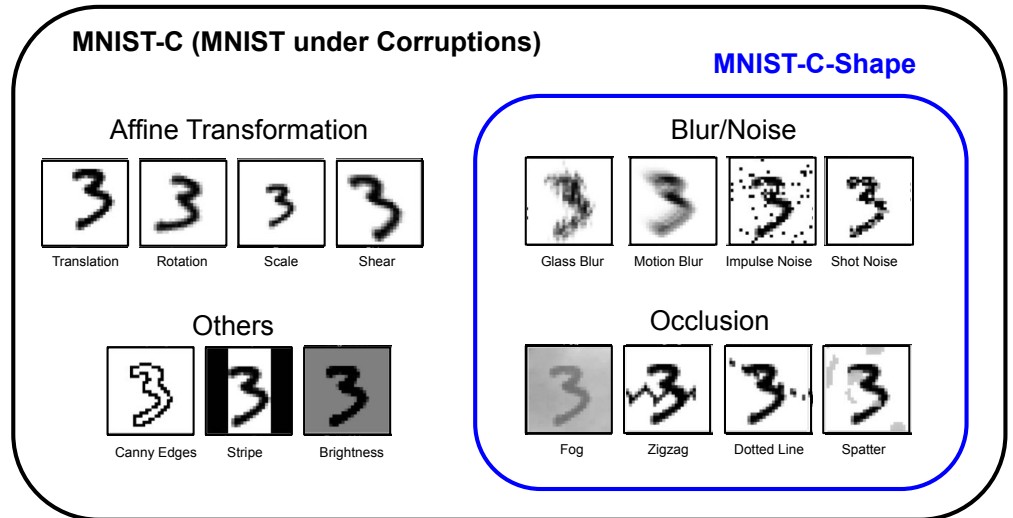

# E   Experiment results on different spatial frequency image reconstructions

We trained our models to reconstruct different spatial frequency components of the input (e.g., reconstruct a low spatial frequency version of an input given a full spectrum image) and tested how it impacts the model performance. We extracted different spatial frequency information from the input using a broadband Gaussian filter. We used cutoff frequency below 6 cycles per image and above 30 cycles per image for generating low and high spatial frequency images, respectively. We found that even the use of a low-resolution object reconstruction yields comparable performance with the model trained with full-spectrum. **Left figure**: Sample images filtered in full-spectrum (top), low (middle), and high spatial frequency (bottom). **Right table**: Average performance results from the models trained to reconstruct either full-spectrum frequency (FS), low-spatial frequency (LSF), or high-spatial frequency (HSF) images. SDs are in parentheses.

|  | FS | LSF | HSF |
| --- | --- | --- | --- |
| MNIST-C | 92.09 (0.28) | 91.84 (0.67) | 91.29 (0.49) |
| MNIST-C-shape | 96.38 (0.45) | 96.24 (0.74) | 94.73 (0.67) |

