# OpenReview forum: "Reconstruction-guided attention improves the robustness and shape processing of neural networks"
_NeurIPS.cc/2022/Workshop/SVRHM — SVRHM Poster_

### Official Review · Reviewer_jQRG · 2022-10-11
**Inspiring ideas and promising results**

**Rating:** 6
**Confidence:** 4

**Review:**

The authors investigate the role of input reconstruction in the robustness of object recognition.
For this purpose, they injected a reconstruction module as a top-down attentional feedback, inspired by similar phenomena in human visual system such as image completion and pareidolia.
The authors demonstrate significantly improved robustness on MNIST-C, and draw parallels to human-like errors when feeding noise in their model.
I find the investigation inspiring and insightful. I also find the background section very informative.

It would be helpful to experiment with datasets beyond MNIST to support the parallels with human vision. ImageNet has a similar variant to test robustness to various transformations.

It would also be helpful demonstrate the architecture on image completion and pareidolia. This does require extensive work beyond the scope and expectation at SVRHM, but would provide a strong case for the working hypothesis and for demonstrating the impact of the proposed addition beyond a few robustness measures.

---

### Official Review · Reviewer_UeBV · 2022-10-12
**Interesting idea and positive results in limited evaluation settings, does not consider prior research along similar lines**

**Rating:** 6
**Confidence:** 4

**Review:**

**Summary:**

The authors propose a novel training mechanism for object detection, which involves adding a feedback channel to a capsule net where the object's spatial mask is learned and then used to guide the object-centric features that are learned by the original capsule net. The authors show that it leads to an improvement in performance on MNIST-C for a two-layer CNN and ResNet18, and also that this pre-training does not cause ResNet to overfit to MNIST in a manner that reduces downstream robustness on out-of-distribution MNIST-C.

**Strengths:**

The paper is well-written and easy to follow. The problem of out-of-distribution generalization is a long-standing problem in object recognition and the authors do a good job at drawing connections to biological studies which explore human object recognition. The idea of using a semantic mask to guid object recognition is not novel, as prior work on multi-task learning has explored this. But using object-centric representations for the reconstruction and added supervision is new. The method reduces texture bias in training a two-layer ConvNet and ResNet18, as demonstrate through better downstream performance on MNIST-C which has several low-level image corruptions such as blur and localized noise.

**Weaknesses:**

The authors suggest that using object-centric representations with a shape bias is better for downstream out-of-distribution robustness quite generally ("Our model showed strong generalization performance against various image perturbations, on average outperforming all other models including feedforward CNNs and adversarially trained networks.") This might be the case when the downstream task is object recognition with low-level noise such as MNIST digits with motion blur. However, there are other downstream tasks such as semantic segmentation or inpainting where low-level features (including texture information) is quite important.

The authors report results on very small models on a very simplistic dataset (low-dimensional, low-class diversity). It is hard to extrapolate from this work whether this method will translate well to real-world object recognition problems.

One of my major concerns with the paper is that it does not consider literature and prior work outside capsule networks and CNNs.

1. The authors should situate this work wrt to the work on masked image modelling using vision transformers (e.g.  Masked autoencoders or MAEs [1]) which explicitly relies on image reconstruction loss as a self-supervised pre-training method of visual representations. Vision transformers utilize multi-head self-attention as their key component, and hence in many ways the masked autoencoder is a reconstruction-guided attention learning mechanism.

The MAE authors also report strong (in many cases state-of-the-art) robustness results on downstream evaluation where they pre-train on ImageNet and test on relatively more semantically complex datasets (compared to MINIST-C) such as ImageNet-Corruption, ImageNet-Rendition,, ImageNet-Rendition, ImageNet-Sketch. It seems remiss to not mention this literature on masked image modelling of vision transformers.

2. The idea of using object-centric slot attention was initially proposed in [2] and has led to several follow-up works in using object-centric visual features in various ways, which includes implications on robustness [3]. The authors do not dicuss what the prior works on using obect-centric representations for robustness have covered.

**Questions and Minor Suggestions:**

1. Figure 1 appears to have been clipped at the top

2. The abbreviation RT in Section 3.2 and Figure 3 has been used without any definition. What does RT refer to? Based on the description and interent search it seems to imply Response Time [4] based on classifier confidence. If so, please cite the relevant literature.


**References**

1. He, K., Chen, X., Xie, S., Li, Y., Dollár, P. and Girshick, R., 2022. Masked autoencoders are scalable vision learners. In Proceedings of the IEEE/CVF Conference on Computer Vision and Pattern Recognition (pp. 16000-16009).
2. Locatello, F., Weissenborn, D., Unterthiner, T., Mahendran, A., Heigold, G., Uszkoreit, J., Dosovitskiy, A. and Kipf, T., 2020. Object-centric learning with slot attention. Advances in Neural Information Processing Systems, 33, pp.11525-11538.
3. Dittadi, A., Papa, S.S., De Vita, M., Schölkopf, B., Winther, O. &amp; Locatello, F.. (2022). Generalization and Robustness Implications in Object-Centric Learning. Proceedings of the 39th International Conference on Machine Learning, in Proceedings of Machine Learning Research 162:5221-5285
4. Taylor, E., Shekhar, S., & Taylor, G. W. (2020). Response time analysis for explainability of visual processing in CNNs. In Proceedings of the IEEE/CVF Conference on Computer Vision and Pattern Recognition Workshops (pp. 382-383).

---

### Official Review · Reviewer_oTzb · 2022-10-13
**Great paper, strong accept**

**Rating:** 9
**Confidence:** 4

**Review:**

This paper introduces a reconstruction-guided attention method to improve the robustness and shape processing of neural networks. They use an architecture with an encoder, feature capsules, object capsules, and separate reconstruction and classification branches. The reconstruction was used to generate a spatial attention mask, whereas the mapping of features to objects served as a form of feature-based attention. These spatial attention mask was applied to the input, and the process iterated, until a decision criterion was met in the classifier. The spatial and feature attention mechanisms improved training efficiency, in increased robustness to image distortions, particularly those putatively pertaining to shape (blur, noise, and occlusion).

The idea and approach appears highly original and quite effective, and the work was presented thoroughly and clearly.

I think it would be worthwhile for the authors to address the relationship between reconstruction-for-attention (their work) versus reconstruction-for-recognition (e.g., Yildirim, Belleonne, et al., 2020), and to discuss possible limitations of their approach (e.g., whether/how it would scale to imagenet, ecoset, or more generally natural image/object recognition). It would also be interesting to understand how this approach to attention relates to Heeger & Reynold’s divisive normalization approach, which provides a unified mechanism for both spatial and feature-based attention (to my knowledge not implemented in deepnets, but could be). Finally, I believe this line of work could benefit from a more stringent or direct test of shape processing, which is perhaps only indirectly related to the image distortions categorized as shape-relevant (i.e., blur, noise, occlusion).

---

### Official Review · Reviewer_NCT2 · 2022-10-16
**Reconstruction-guided attention improves the robustness and shape processing of neural networks**

**Rating:** 7
**Confidence:** 5

**Review:**

Strengths:
- Excellent introduction that covers past work and well motivates the problem of integrated bottom-up information with top-down reconstruction to create a more robust visual recognition system.
- This work proposes a novel training method that iteratively refines an attention mask used to improve recognition robustness based on an object's reconstruction.
- Results demonstrate a clear benefit of the proposed system on MNIST-C classification accuracy.

Weaknesses:
- MNIST-C digits are fairly simple and can be parameterized in a generative model by a relatively small number of features. Will this technique scale to more difficult datasets? I suggest repeating the current experiments with CIFAR10-C from https://github.com/hendrycks/robustness.
- Does adding the decoder to the system (3 fully-connected dense layers) significantly increase the total parameter count? This may account for some of your systems performance benefits and make your comparison to a baseline ResNet-18 and 2 Conv network a bit unfair (although the significant performance gains observed suggest this is likely not the case)

Misc:
- In figure 2a, it looks to me like the reconstruction is actually getting worse, not better, after a couple feedback+masking steps. This “8” image may not be the best example to illustrate your point.
- Figure 2a is easier to interpret for me than figure 1. Consider adding the same terms from figure 2 like feedforward and feedback to figure 1.